# Calcium-Free and Cytochalasin B Treatment Inhibits Blastomere Fusion in 2-Cell Stage Embryos for the Generation of Floxed Mice via Sequential Electroporation

**DOI:** 10.3390/cells9051088

**Published:** 2020-04-28

**Authors:** Takuro Horii, Ryosuke Kobayashi, Mika Kimura, Sumiyo Morita, Izuho Hatada

**Affiliations:** Laboratory of Genome Science, Biosignal Genome Resource Center, Institute for Molecular and Cellular Regulation, Gunma University, 3-39-15 Showa-machi, Maebashi 371-8512, Japan; horii@gunma-u.ac.jp (T.H.); rkobayashi@gunma-u.ac.jp (R.K.); mikimura@gunma-u.ac.jp (M.K.); msumiyo@gunma-u.ac.jp (S.M.)

**Keywords:** CRISPR/Cas9, knock-in, flox, electroporation, 2-cell, fusion

## Abstract

The generation of conditional knockout mice using the Cre-loxP system is advantageous for the functional analysis of genes. Flanked by two loxP sites (floxed) mice can be directly obtained from fertilized eggs by the CRISPR/Cas9 genome editing system. We previously reported that sequential knock-in (KI) of each loxP site by electroporation (EP) at the 1- and 2-cell embryonic stages increases the number of mice with floxed alleles compared with simultaneous KI. However, EP at the 2-cell stage frequently induced blastomere fusion. These fused embryos cannot develop to term because they are tetraploidized. In this study, we examined the following three conditions to inhibit blastomere fusion by EP at the 2-cell stage: (1) hypertonic treatment, (2) Calcium (Ca^2+^)-free treatment, and (3) actin polymerization inhibition. Hypertonic treatment of 2-cell stage embryos prevented blastomere fusion and facilitated blastocyst development; however, KI efficiency was decreased. Ca^2+^-free treatment and actin polymerization inhibition by cytochalasin B (CB) reduced fusion rate, and did not have negative effects on development and KI efficiency. These results suggest that Ca^2+^-free and CB treatment at the 2-cell stage is effective to generate floxed mice in combination with a sequential EP method.

## 1. Introduction

Various genome engineering technologies have been developed to establish gene knockout mice; however, a considerable number lead to embryonic lethal phenotypes [1]. The conditional knockout system allows for spatial and temporal control of genetic modification that can overcome this issue. The Cre-loxP conditional knockout system is the most commonly used. The Cre-loxP method utilizes site-specific Cre recombinase and the loxP site consisting of a unique 34-bp sequence [2]. Cre-mediated recombination occurs by deletion or inversion of a genetic region of interest, flanked by two loxP sites (floxed), leading to targeted gene modification in cells expressing Cre. 

Until recently, the generation of floxed mice was a time-demanding, complicated process, involving homologous recombination in embryonic stem cells (ESCs), chimeric mouse production by microinjection of these ESCs, and the production of heterozygous offspring by crossing chimeric mice with wild-type mice. Advances in genome editing tools, such as the direct injection of engineered endonucleases or RNA-guided nucleases into zygotes, has greatly accelerated the production of gene-modified animals. The most popular system, clustered regularly interspaced short palindromic repeats (CRISPR)/CRISPR-associated 9 (Cas9), is based on RNA-guided nucleases [3]. The CRISPR/Cas9 technology enables the generation of knockout mice by the delivery of Cas9 nuclease and guide RNA (gRNA) to target complementary DNA sequences. Furthermore, co-injection of single- or double-stranded donor DNA templates, homologous to target sequences flanking the double strand break (DSB) site, can facilitate precise point mutations or DNA insertions [4,5,6]. Notably, floxed alleles can be generated by the simultaneous injection of Cas9, two pairs of gRNAs and two single-stranded oligodeoxynucleotides (ssODNs) containing loxP sequences into mouse zygotes [6,7,8,9]. This powerful method simplifies the process by eliminating the need to construct a knock-in (KI) vector; thus, floxed mice can be obtained in a short period of time. However, the method using two pairs of gRNAs and ssODNs causes undesirable chromosomal deletion by inducing two DSBs on the same chromosome, therefore reducing the floxed rate [10,11]. 

To solve this, we developed a method to sequentially introduce each loxP site into the locus of interest at the 1- and 2-cell embryonic stages, respectively (Figure 1) [10]. Sequential electroporation (EP) improved the floxing efficiency compared with ordinary simultaneous microinjection, leading to a high yield of offspring with floxed alleles. However, EP at the 2-cell stage frequently induced blastomere fusion, resulting in tetraploid embryos. Tetraploid embryos arrest development a few days after implantation due to their poor formation of epiblast [12]. Therefore, it is necessary to reduce the rate of blastomere fusion at the 2-cell stage to increase the number of viable embryos. In this study, we examined the following three potential conditions to inhibit blastomere fusion by EP at the 2-cell stage: (1) hypertonic treatment, (2) Ca^2+^-free treatment, and (3) actin polymerization inhibition. This study shows that Ca^2+^-free treatment and inhibition of actin polymerization at the 2-cell stage are effective to reduce blastomere fusion, and maintain embryonic development and KI efficiency.

## 2. Materials and Methods

### 2.1. Mouse

B6D2F1 mice were purchased from CLEA Japan (Kawasaki, Japan). C57BL/6J (B6) and ICR mice were purchased from Charles River Japan (Yokohama, Japan). The mice were maintained on a 12 h light/dark cycle (lights on from 08:00 h to 20:00 h) and were provided with food and water ad libitum. All animal experiments were approved by the Animal Care and Experimentation Committee of Gunma University (No. 17-030), and were carried out in accordance with the approved guidelines. 

### 2.2. Preparation of Embryos

B6D2F1 (8–10 weeks of age) or C57BL/6J (4 weeks of age) female mice were induced to superovulate by injecting 7.5 units of pregnant mare’s serum gonadotropin (PMSG; SEROTROPIN, ASKA Pharmaceutical, Tokyo, Japan) at approximately 16:00 h, followed by the administration of 7.5 units of human chorionic gonadotropin (hCG; GONATROPIN, ASKA Pharmaceutical) 48 h later. After the administration of the hCG, females were mated with male mice of the same strain. Zygotes were isolated from the oviduct 21 h later. After washing in M2 medium (Sigma-Aldrich, St. Louis, MO) including 0.1% (*w*/*v*) bovine hyaluronidase, zygotes were transferred to M16 medium (Sigma-Aldrich) supplemented with penicillin and streptomycin at 37 °C. For the in vitro culture of B6 embryos, M16 medium containing 1 mM EDTA/2Na was used to reverse developmental arrest at the 2-cell stage. EP at the 1-cell stage was conducted at 24–26 h post hCG; EP at the 2-cell stage was conducted at 42–44 h post hCG.

### 2.3. Preparation of Cas9, crRNA, tracrRNA and ssODN

Two CRISPR RNAs (crRNAs) and ssODNs with 5′- and 3′-homology arms (each 60 mer) flanking loxP variants (lox66 or lox71) were designed to target *Mecp2* intron 2 and 3, based on our previous report (Table 1) [10]. To facilitate the detection of correct insertions, the ssODNs were engineered to contain an *Nhe*I restrictionsite and an *Eco*RI restriction site, respectively, in addition to the loxP sequences. Equal volumes of crRNA (100 μM; IDT, Coralville, IA) and trans-activating crRNA (tracrRNA) (100 μM; IDT) were combined in a duplex buffer (IDT), heated in a thermal cycler to 95 °C for 5 min, and then placed for 10 min at room temperature according to the manufacture’s protocol. Combined crRNA/tracrRNA (3 μM) was mixed with recombinant Cas9 protein (100 ng/μL; GeneArt Platinum™ Cas9 Nuclease, Thermo Fisher Scientific, Waltham, MA, USA) and ssODNs (400 ng/μL) in Opti-MEM I (Life Technologies, Carlsbad, CA, USA), and the mixture was placed for 10 min at room temperature before EP.

### 2.4. Electroporation

EP was performed as described previously [10]. In brief, the electrode (LF501PT1–10; BEX, Tokyo, Japan) connected with the CUY21EDIT electroporator (BEX) was set under a stereoscopic microscope. Embryos were washed twice with Opti-MEM I solution and placed in a line in the electrode gap filled with 5 μL Cas9/crRNA/tracrRNA/ssODN mixture. Electroporation was performed using 30 V (3 msec ON + 97 msec OFF) with seven electric pulses. To determine in vitro development and KI efficiency, embryos were cultured in M16 medium at 37 °C until the blastocyst stage. To obtain newborn mice, 2-cell stage embryos were transferred to the oviduct of pseudopregnant ICR females. 

### 2.5. Inhibition of Blastomere Fusion

For hypertonic treatment, a NaCl solution was added to the EP mixture instead of Opti-MEM I. For example, adjusting to +0.1 M, 1 µL 1 M NaCl solution was added to 9 μL Opti-MEM I based isotonic EP buffer. NaCl molarity (*w*/*v*) was adjusted to +1 M (+5.8%), +0.5 M (+2.9%), +0.3 M (+1.8%), +0.2 M (+1.2%) and +0.1 M (+0.6%), compared with the normal EP mixture. For cadherin inhibition, 2-cell embryos were treated with Ca^2+^ and Mg^2+^-free PBS (PBS (−)) or Ca^2+^-free M2 medium [13] for 20 min at 37 °C before (and after) EP. For actin polymerization inhibition, 2-cell stage embryos were cultured in M2 medium containing 5 μg/mL cytochalasin B (CB, Sigma-Aldrich) at 37 °C for 20 min before and/or after EP. After the treatment, embryos were returned to M16 medium and incubated at 37 °C. The number of fused embryos were counted 1 h after incubation in M16 medium.

### 2.6. Assay for KI and Floxed Allele

Genomic DNA of blastocysts and newborn mice was extracted using DirectPCR Lysis Reagent (Viagen Biotech, Los Angeles, CA, USA) according to the manufacture’s protocol. To detect loxP insertion in the target *Mecp2* gene, the following primers were used: for intron 2 (Left PCR), Mecp2loxPCheck-1 (5′-AAGAAGCCAACCATACAGTGC-3′) and Mecp2loxPCheck-3 (5′-TGAGTGCCACACATGAGACC-3′); for intron 3 (Right PCR), Mecp2loxPCheck-4 (5′-GGGTAGGAAGGCTAGGATGG-3′) and Mecp2loxPCheck-2 (5′- GCTTGCTCAGAAGCCAAAAC-3′); for the floxed allele insertion (Long PCR), Mecp2loxPCheck-1 (5′-AAGAAGCCAACCATACAGTGC-3′) and Mecp2loxPCheck-4 (5′- GGGTAGGAAGGCTAGGATGG-3′). The expected sizes of the PCR products of the wild-type allele for the Left, Right, and Long PCR were 238, 276, and 983 bp, respectively.

PCR was performed in a 10 μL reaction volume. The PCR conditions were as follows: initial denaturation (94 °C for 1 min); followed by 40 cycles (for blastocysts) or 35 cycles (for newborn mice) for denaturation (94 °C for 10 s), annealing (60 °C for 30 s), and polymerization (72 °C for 1 min); and a final extension at 72 °C for 5 min using rTaq DNA polymerase (TaKaRa Taq; #R001A, Takara Shuzo, Shiga, Japan). To detect KI efficiency, PCR products were digested with *Nhe*I and/or *Eco*RI enzymes, which cleave inserted alleles including loxP sites. The PCR products and digested samples were resolved on a 2% agarose gel and visualized by staining with ethidium bromide. Alternatively, samples were analyzed using capillary and microchip electrophoresis (MCE-202 MultiNA; Shimadzu, Kyoto, Japan).

### 2.7. Statistical Analysis

The diameter of the blastomeres and the radius (*r*) of the adhesive area were measured using ImageJ software (NIH). The adhesive area (µm^2^) between blastomeres was calculated using a following formula: 3.14 × *r^2^*. Student’s *t*-test (two-tailed test) was applied to calculate *p* values for them. Fisher’s exact probability test was used to calculate *p* values when comparing fusion, burst, development, born and KI efficiency. A *p*-value of < 0.05 was considered significant.

## 3. Results

### 3.1. Experiment 1: Hypertonic Treatment

Sequential EP is generally performed under isotonic conditions. Previous reports, aimed at hybridoma production, demonstrated that EP performed under hypotonic conditions increases the rate of cell fusion [14,15]. This effect could be attributed to the improved physical contact between adjacent cells, or the enhanced fusogenic state of the cell membrane. We hypothesize that EP inhibits cell fusion when performed under hypertonic conditions (Figure 2A). We examined five hypertonic conditions (+1 M, +0.5 M, +0.3 M, +0.2 M, and +0.1 M NaCl) by the addition of a NaCl buffer to the EP mixture. Under all hypertonic conditions, the blastomeres were observed to shrink, and the cytoplasm became dense and concentrated (Figure 2B and Appendix A). The blastomeres reduced in size depending on the tonicity of the EP buffer (Figure 2C). Preliminary experiments of EP at the 2-cell stage (2-cell EP) were performed with a constant voltage (30 V). In this condition, about 20% of 2-cell stage embryos were fused by electric stimulation. As hypothesized, hypertonic conditions containing a minimum of +0.3 M NaCl inhibited electric fusion between two blastomeres (Figure 2D). By contrast, the most hypertonic condition (+1 M NaCl) reduced blastocyst formation (Figure 2D and Appendix A). The current value under the constant voltage condition is positively correlated with the concentration of NaCl, which is added to the EP buffer. The +1 M NaCl condition increased the electric current compared with the control (0.61 A vs. 0.17 A), causing damage and reduced embryonic development. To examine EP under constant current (0.20 A), the voltage was altered. Under constant current and hypertonic conditions, electric fusion was completely inhibited in all hypertonic conditions (Figure 2E) and blastocyst formation was significantly improved compared with the control (Figure 2E and Appendix A). 

Next, the effects of hypertonic treatment were tested with a sequential EP procedure. The first EP was performed under isotonic conditions at the 1-cell stage, followed by a second EP at the 2-cell stage under hypertonic (+0.1 M NaCl) or normal isotonic conditions. Under hypertonic conditions, blastomere fusion was inhibited, facilitating blastocyst formation (Figure 2F). However, KI efficiency was significantly decreased in the second EP step, indicating a decreased floxed rate compared with control conditions (Figure 2G). Thus, sequential EP combined with hypertonic treatment is unsuitable for efficient generation of floxed mice.

### 3.2. Experiment 2: Ca^2+^-Free Treatment

Cadherins are a family of cell adhesion receptors that are crucial for the mutual association of vertebrate cells [16]. E-cadherin is essential for the development of preimplanted mouse embryos. E-cadherin null embryos show severe developmental abnormalities at the morula and blastocyst stages [17,18,19]. In particular, the adhesive cells of the morula dissociate shortly after compaction has occurred, destroying their morphological polarization. E-cadherin null embryos at the 2-cell stage show normal morphology; however, individual blastomeres dissociate after the removal of the zona pellucida [19]. Therefore, temporary inhibition of E-cadherin in electroporated 2-cell stage embryos could repress blastomere adhesion and lead to a reduced fusion rate (Figure 3A). The cadherins are transmembrane glycoproteins that play a key role in Ca^2+^-dependent cell–cell adhesion, and the 2-cell stage embryo loses blastomere adhesion in the absence of Ca^2+^ [20]. According to this report, 5–45 min of Ca^2+^-free treatment is sufficient for the loss of blastomere adhesion. Based on this, 2-cell stage embryos were treated with Ca^2+^-free PBS (PBS(−)) for 20 min. After a treatment of PBS(−), the two blastomeres were observed to dissociate (Figure 3B) and the adhesive area between blastomeres was significantly reduced (Figure 3C). All EP experiments were conducted with the 2-cell stage embryo (2-cell EP). PBS(−) treatment was applied either before EP (−/+) or spanning EP (−/−) (Figure 3D). Almost all embryos (−/+) were viable, whereas 48% of embryos (−/−) burst during PBS (−) treatment after EP and were not viable (Appendix A). The blastomere fusion rate was significantly reduced by the PBS(−) treatment (Appendix A), but the developmental rate of surviving diploid embryos declined (Appendix A). Therefore, PBS(−) treatment is unsuitable for inhibiting fusion in 2-cell stage embryos. 

Next, we examined Ca^2+^-free M2 embryo culture medium, with more enriched medium composition than PBS(−), to inhibit blastomere fusion. As expected, Ca^2+^-free M2 treatment also inhibited blastomere fusion following EP (Figure 3E,F). The fusion rate of blastomeres was 22% in control embryos, compared with 12% in embryos (−/+) and 11% in embryos (−/−). This result indicates that Ca^2+^-free treatment before EP is necessary to inhibit blastomere fusion. Fortunately, embryos treated with Ca^2+^-free M2 did not show a reduction in viability, development and KI rates (Figure 3E). To investigate the effects of Ca^2+^-free treatment on flox efficiency, sequential EP was performed. The first EP was performed at the 1-cell stage under standard conditions; the second EP at the 2-cell stage was performed under Ca^2+^-free or standard conditions. It was observed that Ca^2+^-free treatment before EP significantly inhibited cell fusion at the 2-cell stage (Figure 3G). Importantly, Ca^2+^-free treatment did not have negative effects on blastocyst development and flox efficiency after sequential EP (Figure 3G,H).

### 3.3. Experiment 3: Actin Polymerization Inhibition

Actin protein is present as either a free monomer called G-actin or as part of a linear polymer microfilament called F-actin, both of which are essential for cellular movement and adhesion during cell division. The actin polymerization inhibitor, cytochalasin, causes a decrease in F-actin and an increase in G-actin content [21]. It has previously been reported that cytochalasin inhibits blastomere adhesion at the cleavage and morula stages [22,23]. To examine whether the inhibition of actin polymerization had an effect on blastomere fusion, cytochalasin B (CB) was applied to 2-cell EP embryos before and/or after EP (Figure 4A). In brief, CB treatment was conducted for 20 min before EP (CB+/CB−), 20 min after EP (CB−/CB+), or 20 min before and after EP (CB+/CB+). The embryos treated with CB demonstrated no apparent morphological change compared with control embryos (Figure 4B). However, CB treatment significantly inhibited blastomere fusion following EP (Figure 4C,D). The fusion rate of blastomeres was 23% in the control and 25% in CB−/CB+ embryos, compared with 10% in CB+/CB− and 3% in CB+/CB+ embryos. This result inidicates that CB treatment spanning EP (CB+/CB+) is the most effective to inhibit blastomere fusion. In contrast, we expected the combined effect of Ca^2+^-free and CB treatment on blastomere fusion, but this did not have a better effect than the independent treatment of Ca^2+^-free or CB (Appendix A).

To investigate the effects of CB treatment on flox efficiency, sequential EP was performed. The first EP was performed at the 1-cell stage under standard CB absent (CB−) conditions; the second EP at the 2-cell stage was performed under CB+/CB−, CB+/CB+, or control (CB−/CB−) conditions. It was observed that CB treatment significantly inhibited cell fusion at the 2-cell stage, particularly in the CB+/CB+ condition (Figure 4E). Importantly, CB treatment did not have negative effects on blastocyst development and KI efficiency after sequential EP (Figure 4E). To examine the in vivo development of CB− treated embryos, floxed mice were generated by CB+/CB+ treatment or control (CB−/CB−). Confirming the in vitro results, blastomere fusion of the 2-cell stage embryo was significantly inhibited in the CB+/CB+ condition compared with the control (Figure 5A,B). CB treatment did not have negative effects on in vivo development, KI or flox efficiency (Figure 5A,B). Furthermore, the optimized method was successful in both B6D2F1 and B6 mouse strains. These data suggest that actin polymerization inhibition can successfully reduce the rate of blastomere fusion in 2-cell EP embryos without any negative effects.

## 4. Discussion

In this study, the following three conditions were examined to inhibit blastomere fusion by EP at the 2-cell embryonic stage: (1) hypertonic treatment, (2) Ca^2+^-free treatment and (3) actin polymerization inhibition. 

In Experiment 1, hypertonic treatment of 2-cell stage embryos inhibited blastomere fusion due to repressed cell adherence or the degraded fusogenic state of the cell membrane. The hypertonic condition improved blastocyst development and reduced KI efficiency, probably via a reduction in blastomere volume caused by water moving out of the cell to compensate for increased osmotic pressure. Therefore, the total uptake of Cas9/crRNA/tracrRNA/ssODN would decrease due to reduced blastomere volume. Furthermore, ssODN is toxic to the embryo; therefore, the concentration of ssODN delivered by EP would directly affect embryonic development rate [10]. Thus, when the ssODN uptake is decreased under hypertonic conditions, it is expected that the developmental rate will increase and KI efficiency will decrease. Alternatively, cell membrane permeability was potentially degraded by hypertonic treatment, leading to a decrease in Cas9/crRNA/tracrRNA/ssODN uptake following EP. In support of this hypothesis, propidium iodide (PI) uptake by EP in B16-F1 and CHO cell lines was higher in the hypotonic versus the isotonic buffer [15]. Otherwise, EP, especially sequential EP, under isotonic conditions may damage the embryos. Hypertonic treatment before and after EP will reduce embryo damage but lead to a reduced KI efficiency.

Ca^2+^-free treatment is an established method to reduce cell adhesion. In Experiment 2, PBS(−) treatment partially inhibited cell fusion; however, 2-cell stage embryos had reduced viability and development. PBS(−) treatment after EP was detrimental to embryo viability and may be toxic to cell membrane recovery from electric damage. By contrast, Ca^2+^-free M2 treatment also inhibited cell fusion and did not have negative effects on in vitro development and KI rate. To efficiently inhibit cell fusion, a combination of Ca^2+^-free buffer and another cell adherence inhibitor, such as Ethylenediaminetetraacetic acid (EDTA), known as Ca^2+^ chelating agent or Cadherin inhibitor(s), may be effective and could be investigated further.

In Experiment 3, blastomere fusion rate was investigated after actin polymerization inhibition. Embryos at the cleavage-stage have an abundance of actin immediately beneath the cell membrane, which is occasionally located inter-cellularly [24,25]. Previous studies indicate that the actin polymerization inhibitor, cytochalasin, can prevent blastomere adhesion in preimplantation embryos. De-compaction of morula-stage embryos can be induced by treatment with cytochalasin [23]. Furthermore, 2-cell stage embryos treated with cytochalasin for 44–48 h have well-separated blastomeres, indicating a loss of adhesion [22]. Long-term treatment with cytochalasin usually induces polyploidy because cytochalasin allows chromosomal replication to proceed while arresting cytoplasmic division. Therefore, short-time treatment (20 min) of CB was tested in this experiment. Despite no observable morphological changes, cell fusion was significantly inhibited by CB treatment before and after EP. Further, CB treatment had no effect on either in vitro or in vivo development, nor on KI efficiency. The blastomere fusion cannot be completely suppressed by CB treatment; therefore, we expected the combined effect of Ca^2+^-free and CB treatment on blastomere fusion. However, this did not have a better effect than the independent treatment of Ca^2+^-free or CB treatment. CB treatment time or other actin polymerization inhibitors (e.g., Latrunculin A) can be further optimized and warrant further testing. Theoretically, CB treatment could achieve a 10% increase in floxed mice as a result of a 10% reduction in blastomere fusion. Together, these data suggest that Ca^2+^-free treatment and the inhibition of actin polymerization at the 2-cell stage can generate floxed mice by a sequential EP method.

In recent years, various methods have been reported as potential alternatives to the two-donor floxing technique. Methods using long single-stranded DNA (lssDNA) as donors are known as *Easi-*CRISPR (efficient additions with ssDNA inserts-CRISPR) [26], or CLICK (CRISPR with lssDNA inducing cKO alleles) [27]. Both techniques introduce lssDNA as follows: *Easi*-CRISPR is a microinjection-based approach, whereas CLICK is an EP-based approach. Alternatively, Tild-CRISPR (targeted integration with linearized dsDNA-CRISPR) uses long-dsDNA as donors [28]. These newer techniques achieve an improved KI efficiency compared with the simultaneous introduction of two pairs of ssODNs and gRNAs [11]. Unlike the two-donor floxing technique, the new methods are limited by the lengths of the homology arms and of the loxP flanked region. The sequential KI method improved the flox rate compared with simultaneous KI. Using sequential EP, floxed mice were successfully generated with more than 20 kb of flanking region by two loxP sites (data not shown). Recently, a new approach was reported called sequential *i*-GONAD (*si*-GONAD) [29], which sequentially electroporates two gRNAs (and two ssODNs) via oviductal nucleic acid delivery. Although the sequential EP method requires further improvement for not only the inhibition of blastomere fusion at 2-cell stage, but also improvement of embryonic development and successful KI, it has great potential alongside other floxing methods.

## 5. Conclusions

Sequential EP of loxP donors in 1- or 2-cell stage embryos is more efficient than simultaneous EP for the generation of floxed mice. However, a limitation of EP at the 2-cell stage is the frequency of blastomere fusion, which is embryonic lethal because of tetraploidy. Optimization of the sequential EP method revealed that Ca^2+^-free treatment and actin polymerization inhibition effectively reduced fusion rate, thereby increasing the number of viable embryos to obtain more floxed mice.

## Figures and Tables

**Figure 1 cells-09-01088-f001:**
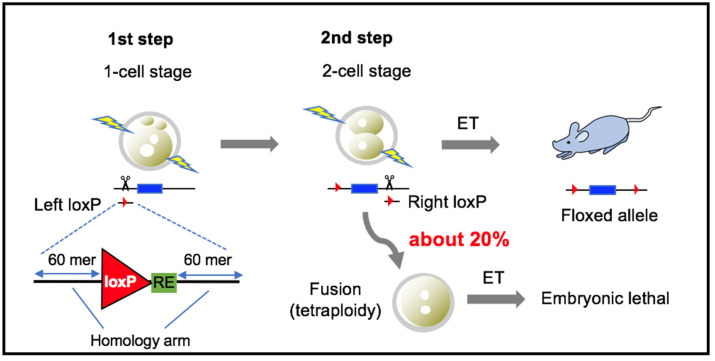
Schematic of sequential EP procedure. At the 1-cell stage, one loxP sequence is inserted by EP. At the 2-cell stage, the other loxP sequence is inserted by EP. Blastomere fusion occurred in 20%, reducing the number of available embryos. RE = restriction enzyme recognition site, ET = embryo transfer.

**Figure 2 cells-09-01088-f002:**
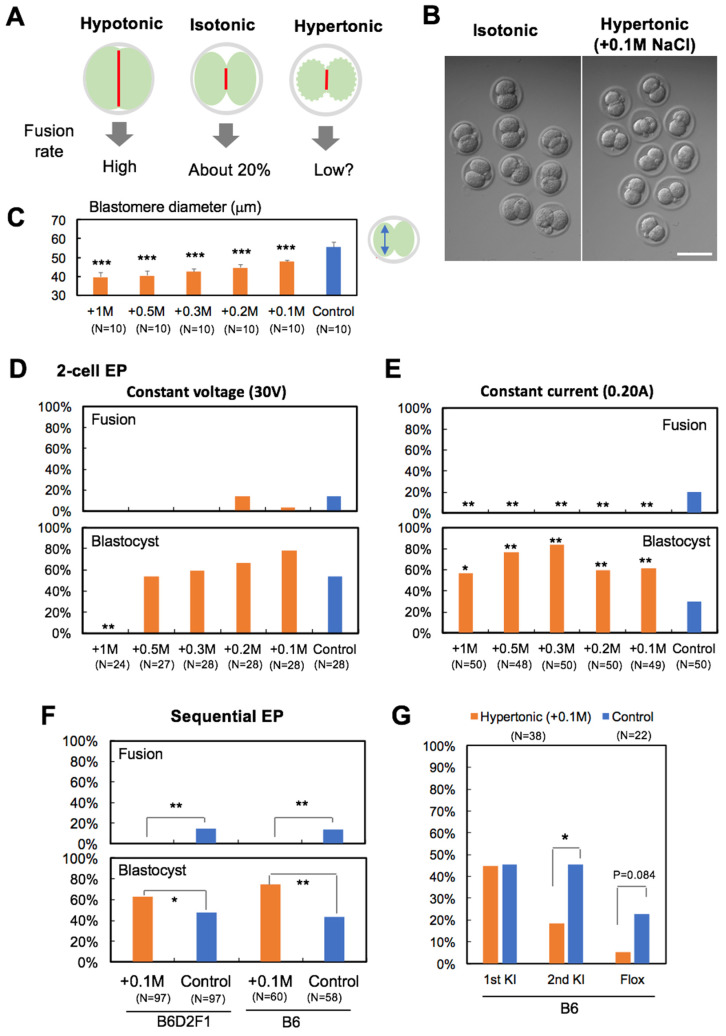
Effects of hypertonic treatment on blastomere fusion, embryonic development and KI efficiency of 2-cell stage embryos. (**A**) Expected relationship between tonicity and cell fusion. (**B**) Morphology of 2-cell stage embryos (B6D2F1) after 20 min treatment of isotonic or hypertonic EP buffer. (**C**) Blastomere diameter under hypertonic conditions. (**D**) Fusion and development (blastocyst) rate of 2-cell EP embryos under constant voltage. (**E**) EP under constant current. (**F**) Fusion and development (blastocyst) rate of 2-cell stage embryos after sequential EP under constant current. (**G**) KI efficiency of blastocysts by sequential EP under hypertonic or isotonic conditions. N indicates the number of 2-cell stage embryos (**D**–**F**) and blastocysts (**G**) used for the analysis. * *p* < 0.05, ** *p* < 0.01, *** *p* < 0.001 (compared with control). Scale bar, 100 μm.

**Figure 3 cells-09-01088-f003:**
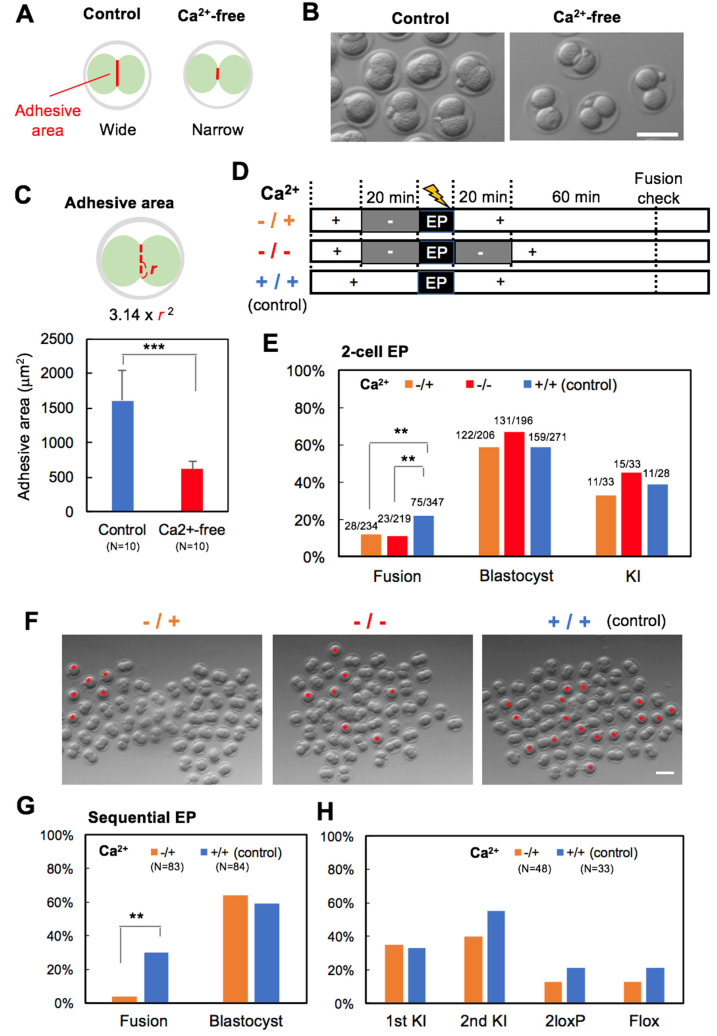
Effects of Ca^2+^-free buffer treatment on blastomere fusion and embryonic development of 2-cell EP. (**A**) Relationship between adhesive area and Cadherin inhibition by Ca^2+^-free treatment. (**B**) Morphology of 2-cell stage embryos (B6D2F1) after 20 min treatment of Ca^2+^-free buffer (PBS). (**C**) Adhesive area beween blastomeres. (**D**) Experimental timeline of Ca^2+^-free treatment. (**E**) Fusion, development (blastocyst) and KI rate by 2-cell EP after Ca^2+^-free M2 treatment. (**F**) Morphology of 2-cell EP embryos. Asterisks indicate fused embryos. (**G**) Fusion and development (blastocyst) rate of 2-cell stage embryos after sequential EP. (**H**) KI efficiency of blastocysts by sequential EP. 2loxP = embryos with two loxP sites in *cis* or *trans*; flox = emryos with two loxP sites in *cis*. N indicates the number of 2-cell stage embryos (**G**) and blastocysts (**H**) used for the analysis. ** *p* < 0.01, *** *p* < 0.001 (compared with the control). Scale bars, 100 μm.

**Figure 4 cells-09-01088-f004:**
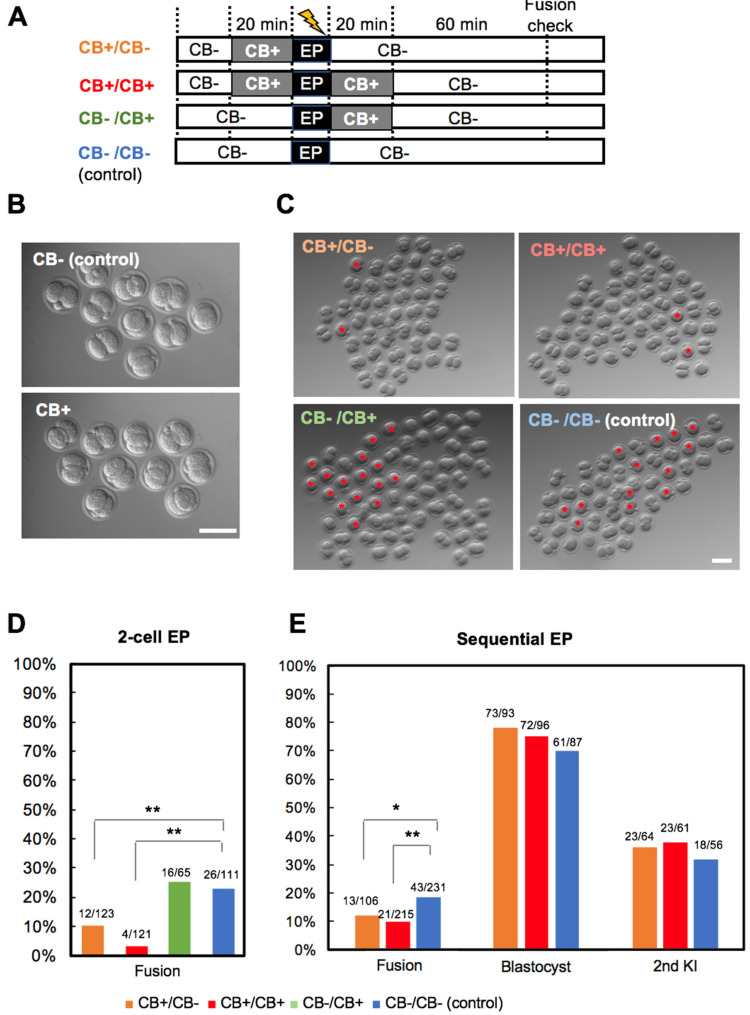
Effects of CB treatment on blastomere fusion, embryonic development and KI efficiency in vitro. (**A**) Experimental timeline of CB treatment. (**B**) Morphology of 2-cell stage embryos (B6D2F1) after 20 min treatment of CB (5 µg/mL). (**C**) Morphology of 2-cell EP embryos. Asterisks indicate fused embryos. (**D**) Fusion rate of 2-cell EP embryos. (**E**) Fusion rate, development (blastocyst) rate, and KI efficiency after sequential EP. * *p* < 0.05, ** *p* < 0.01. Scale bars, 100 µm.

**Figure 5 cells-09-01088-f005:**
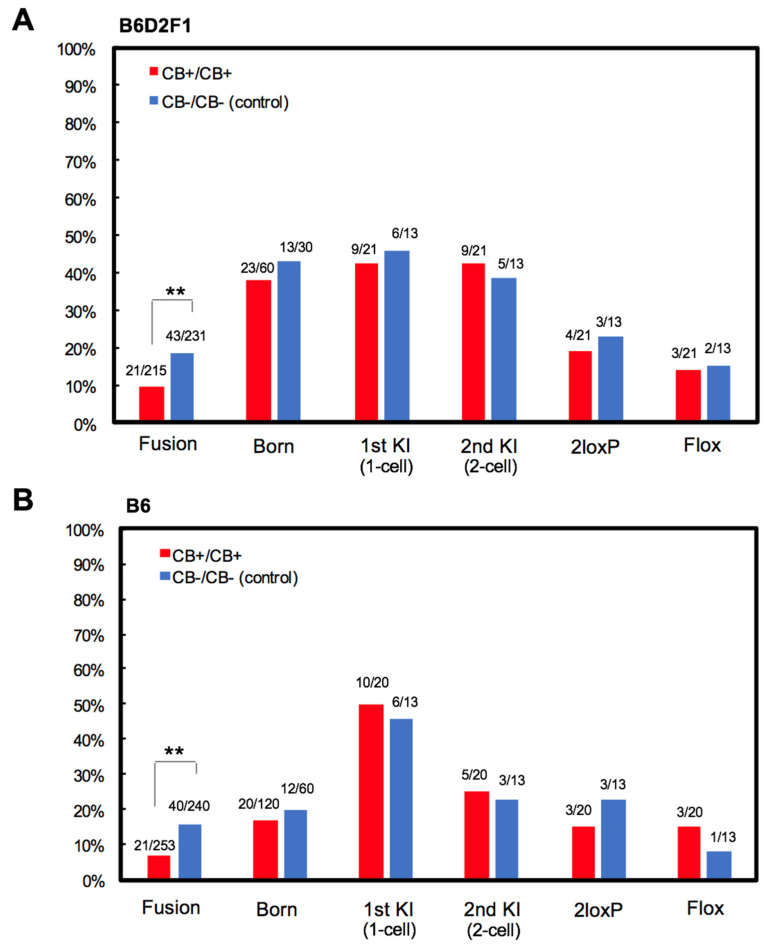
Effects of CB treatment on 2-cell embryos treated with sequential EP in vivo. Fusion rate, birth rate (born/ET), and KI efficiency after sequential EP were measured in both (**A**) B6D2F1 and (**B**) B6 mice. 2loxP = mice with two loxP sites in cis or trans; flox = mice with two loxP sites in cis. ** *p* < 0.01.

**Table 1 cells-09-01088-t001:** crRNA and ssODN used in this study.

Locus	crRNA	ssODN
*Mecp2* intron 2 (Left)	Mecp2-L2 (5′-CCCAAGGATACAGTATCCTA-3′)	Mecp2-L2-lox66 (5′-ccagcaacctaaagctgttaagaaatctttgggccccagcttgacccaaggatacagtatgctagcTACCGTTCGTATAATGTATGCTATACGAAGTTATCCTAGGgaagttaccaaaatcagagatagtatgcagcagccaggggtctcatgtgtggca-3′)
*Mecp2* intron 3 (Right)	Mecp2-R1 (5′-AGGAGTGAGGTCTAGTACTT-3′)	Mecp2-R1-lox71 (5′-ccactcctctgtactccctggcttttccacaatccttaaactgaaggagtgaggtctagtTACCGTTCGTATAGCATACATTATACGAAGTTATGAATTCacttgggggtcattgggctagactgaata tctttggttggtacccagacctaatccacca-3′)

Homology arms (lowercase), Restriction enzyme recognition sites (underlined).

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
