# Peer review of "Calcium-Free and Cytochalasin B Treatment Inhibits Blastomere Fusion in 2-Cell Stage Embryos for the Generation of Floxed Mice via Sequential Electroporation"

_cells, 2020, doi:10.3390/cells9051088_

Round 1

Reviewer 1 Report

The manuscript by Horii T et al describes methods to limit blastomere fusion in 2-cell stage embryos during sequential electroporation. The authors describe three methods and show that inhibition of actin filaments significantly reduce blastomere fusion during the sequential electroporation. The results are intriguing and could be used in the field robustly. However, there are a few concerns which when addressed will only enhance the quality of the manuscript presented:

  1. Fig 2B needs to be quantified and described on the basis of some sort of a phenotype. It seems the blastomeres do reduce in size slightly.
  2. Number of embryos for Fig 2C and Fig 2D needs to be mentioned.
  3. Fig 3B needs to be well documented for and quantified.
  4. It is interesting that the bars in Fig 3D do not add up to 100 between conditions? Can the authors explain?
  5. While it is interesting that inhibition of actin dynamics results in inhibition of the fusion, it is not apparent from the studies at what point actin filaments do play a part. To resolve this the authors, need to additionally include CB treatment after electroporation to understand this.
  6. Additionally, it will be prudent to use Latrunculin A to inhibit actin filament assembly to show similar results as that with CB administration.
  7. Does the combined effect of calcium sequestration and actin filament inhibition have a better effect in reducing the blastomere fusion?

Author Response

Reviewer1

The manuscript by Horii T et al describes methods to limit blastomere fusion in 2-cell stage embryos during sequential electroporation. The authors describe three methods and show that inhibition of actin filaments significantly reduce blastomere fusion during the sequential electroporation. The results are intriguing and could be used in the field robustly. However, there are a few concerns which when addressed will only enhance the quality of the manuscript presented:

Response: We greatly appreciate the reviewer’s insightful comments about our study, which have helped us to significantly improve our paper.

Comment: Fig 2B needs to be quantified and described on the basis of some sort of a phenotype. It seems the blastomeres do reduce in size slightly.

Response: Thank you for the insightful comment. We have measured the diameter of blastomere in each condition using ImageJ software (NIH). Indeed, the blastomeres reduced in size depending on the tonicity of EP buffer. We added the data (Fig. 2C) and description (L141L142 and L155–L156).

Comment:  Number of embryos for Fig 2C and Fig 2D needs to be mentioned.

Response: We showed the number of 2-cell stage embryos under the X axis for Fig. 2D (former 2C) and Fig. 2E (former 2D).

Comment: Fig 3B needs to be well documented for and quantified.

Response: Thank you for the helpful comment. We have measured the radius (r) of adhesive area using ImageJ software (NIH), and then the adhesive area between blastomeres was calculated by the formula: 3.14 x r2. The adhesive area between blastomeres was significantly reduced under the calcium-free condition. We added the data (Fig. 3C) and description (L141L143 and L188–L189).

Comment: It is interesting that the bars in Fig 3D do not add up to 100 between conditions? Can the authors explain?

Response:  Fig. 3D (newer Fig. S2A) does not include the number of embryos that stop at 2-cell to morula stages. Therefore, the total number of fusion, burst and development (blastocyst) in each condition does not reach to 100%.

Comment: While it is interesting that inhibition of actin dynamics results in inhibition of the fusion, it is not apparent from the studies at what point actin filaments do play a part. To resolve this the authors, need to additionally include CB treatment after electroporation to understand this.

Response: We agree with the reviewer’s comment. We have performed the experiment of CB treatment after EP (CB-/CB+) (Fig. 4C, D). The fusion rate of blastomeres was  25% in CB-/CB+ embryos, compared to 23% in control (CB-/CB-). These results suggest that the CB treatment after EP is not critical. By contrast, the fusion rate of blastomeres was 10% in CB+/CB- and 3% in CB+/CB+ embryos, indicating that CB treatment spanning EP could be more effective than that before EP.

Comment: Additionally, it will be prudent to use Latrunculin A to inhibit actin filament assembly to show similar results as that with CB administration.

Response:  Thank you for the helpful suggestion. We described Latrunculin A as one of candidate inhibitors of cell fusion (L308–L309). We will examine that in the future.

Comment: Does the combined effect of calcium sequestration and actin filament inhibition have a better effect in reducing the blastomere fusion?

Response:  We agree with the reviewer’s suggestion. Therefore, we have performed the experiment treated with both calcium sequestration and actin filament inhibition under the optimized timeline. In brief, both calcium-free and CB treatment was conducted before EP, and then CB treatment was conducted after EP. However, improvement of blastomere inhibition was not observed (Fig. S3). We have added the description (L242–L244 & L306–L308).

Reviewer 2 Report

The study described by Horii et al.(Manuscript ID: cells-770987) aims to improve the efficiency of the sequential gene-editing method to generate the floxed mice previously reported by the authors; a problem with the previously reported method is that electroporation treatment with 2-cell embryos can cause the cell-fusion of each cleavage cells, resulting in a decrease of the number of full-term developable embryos. The author attempted to compare with each experimental condition to inhibit 2-cell cleavage fusion by electroporation and found that transient inhibition of actin polymerization by cytochalasin B attenuated the fusion. Moreover, they show that the treated embryos can be generated at full term and can be used for floxed mouse generation.
The findings of this study can be applied not only to the generation of floxed mice validated in this manuscript but also to various experiments utilizing gene transfer into 2-cell embryos by electroporation. Although the cytochalasin B treatment cannot be said that fusion was completely suppressed, this study is expected to lead to the determination of the conditions with higher accuracy.
Therefore, this review concluded that this article appears to meet the standards for publication in Cells, contingent on addressing the following minor issues:

1. A more detailed description of the conditions for adjusting the electroporation solution is required because it would affect the resistance parameter of the electroporation. Was 1x or a higher concentration of OptiMEM used for the mixture?
2. It would be helpful for readers if the descriptions of the nucleotide base length of the homology arms of the ssODN are added.
3. This study shows that the calcium-free treatment caused a decrease in the developmental competence of preimplantation development. Is this due to the effects of exposure to PBS, as well as the effects of calcium-free?

Author Response

Reviewer2

The study described by Horii et al.(Manuscript ID: cells-770987) aims to improve the efficiency of the sequential gene-editing method to generate the floxed mice previously reported by the authors; a problem with the previously reported method is that electroporation treatment with 2-cell embryos can cause the cell-fusion of each cleavage cells, resulting in a decrease of the number of full-term developable embryos. The author attempted to compare with each experimental condition to inhibit 2-cell cleavage fusion by electroporation and found that transient inhibition of actin polymerization by cytochalasin B attenuated the fusion. Moreover, they show that the treated embryos can be generated at full term and can be used for floxed mouse generation.

The findings of this study can be applied not only to the generation of floxed mice validated in this manuscript but also to various experiments utilizing gene transfer into 2-cell embryos by electroporation. Although the cytochalasin B treatment cannot be said that fusion was completely suppressed, this study is expected to lead to the determination of the conditions with higher accuracy.

Therefore, this review concluded that this article appears to meet the standards for publication in Cells, contingent on addressing the following minor issues:

Response: We greatly appreciate the reviewer’s insightful comments about our study, which have helped us to significantly improve our paper.

Comment 1: A more detailed description of the conditions for adjusting the electroporation solution is required because it would affect the resistance parameter of the electroporation. Was 1x or a higher concentration of OptiMEM used for the mixture?

Response: We sincerely apologize for any confusion. We used 1x concentration of OptiMEM. For example, adjusting to +0.1M, 1ml of 1M NaCl solution was added to the 9 ml Opti-MEM I based isotonic EP buffer. We described it in the Materials and methods section (L112L113).

Comment 2: It would be helpful for readers if the descriptions of the nucleotide base length of the homology arms of the ssODN are added.

Response: Thank you very much for the helpful comment. We have added the descriptions of the nucleotide base length of the homology arms in Fig. 1, Table 1 and Materials & Methods section (L89–L90 and L101).

Comment 3: This study shows that the calcium-free treatment caused a decrease in the developmental competence of preimplantation development. Is this due to the effects of exposure to PBS, as well as the effects of calcium-free?

Response: We greatly appreciate the reviewer’s insightful comments about this. We have examined the treatment of calcium-free M2 embryo culture medium, with more enriched medium composition than PBS(-), to inhibit blastomere fusion. Surprisingly, the calcium-free M2 treatment did not cause decrease in viability and development, indicating exposure to PBS but not calcium-free cause detrimental effect to embryos. Therefore, we additionally examined flox efficiency by sequential EP using calcium-free M2 medium. Finally, we found that calcium-free M2 treatment inhibit blastomere fusion and did not have negative effects on development and flox efficiency. We have added the data (Fig. 3E-H) and description (L196–L207). We have changed the title of article to “Calcium-free or cytochalasin B treatment inhibits blastomere fusion in 2-cell stage embryos for generation of floxed mice via sequential electroporation” according to the result. Thank you again for the editor’s kind suggestion which significantly improved our paper.